# "How hard is my MDP?"
# The distribution-norm to the rescue

**Odalric-Ambrym Maillard**
The Technion, Haifa, Israel
odalric-ambrym.maillard@ens-cachan.org

**Timothy A. Mann**
The Technion, Haifa, Israel
mann.timothy@gmail.com

**Shie Mannor**
The Technion, Haifa, Israel
shie@ee.technion.ac.il

## Abstract

In Reinforcement Learning (RL), state-of-the-art algorithms require a large number of samples per state-action pair to estimate the transition kernel $p$. In many problems, a good approximation of $p$ is not needed. For instance, if from one state-action pair $(s, a)$, one can only transit to states with the same value, learning $p(\cdot|s, a)$ accurately is irrelevant (only its support matters). This paper aims at capturing such behavior by defining a novel hardness measure for Markov Decision Processes (MDPs) based on what we call the *distribution-norm*. The distribution-norm w.r.t. a measure $\nu$ is defined on zero $\nu$-mean functions $f$ by the standard variation of $f$ with respect to $\nu$. We first provide a concentration inequality for the dual of the distribution-norm. This allows us to replace the problem-free, loose $||\cdot||_1$ concentration inequalities used in most previous analysis of RL algorithms, with a tighter problem-dependent hardness measure. We then show that several common RL benchmarks have low hardness when measured using the new norm. The distribution-norm captures finer properties than the number of states or the diameter and can be used to assess the difficulty of MDPs.

## 1  Introduction

The motivation for this paper started with a question: Why are the number of samples needed for Reinforcement Learning (RL) in practice so much smaller than those given by theory? Can we improve this? In Markov Decision Processes (MDPs, Puterman (1994)), when the performance is measured by (1) the sample complexity (Kearns and Singh, 2002; Kakade, 2003; Strehl and Littman, 2008; Szita and Szepesvári, 2010) or (2) the regret (Bartlett and Tewari, 2009; Jaksch, 2010; Ortner, 2012), algorithms have been developed that achieve provably near-optimal performance. Despite this, one can often solve MDPs in practice with far less samples than required by current theory. One possible reason for this disconnect between theory and practice is because the analysis of RL algorithms has focused on bounds that hold for the most difficult MDPs. While it is interesting to know how an RL algorithm will perform for the hardest MDPs, most MDPs we want to solve in practice are far from pathological. Thus, we want algorithms (and analysis) that perform appropriately with respect to the hardness of the MDP it is facing.

A natural way to fill this gap is to formalize a "hardness" metric for MDPs and show that MDPs from the literature that were solved with few samples are not "hard" according to this metric. For finite-state MDPs, usual metrics appearing in performance bounds of MDPs include the number of states and actions, the maximum of the value function in the discounted setting, and the diameter or sometimes the span of the bias function in the undiscounted setting. They only capture limited properties of the MDP. Our goal in this paper is to propose a more refined notion of hardness.

**Previous work** Despite the rich literature on MDPs, there has been surprisingly little work on metrics capturing the difficulty of learning MDPs. In Jaksch (2010), the authors introduce the **UCRL** algorithm for undiscounted MDPs, whose regret scales with the diameter $D$ of the MDP, a quantity that captures the time to reach any state from any other. In Bartlett and Tewari (2009), the authors modify **UCRL** to achieve regret that scales with the span of the bias function, which can be arbitrarily smaller than $D$. The resulting algorithm, **REGAL** achieves smaller regret, but it is an open question whether the algorithm can be implemented. Closely related to our proposed solution, in Filippi et al. (2010) the authors provide a modified version of **UCRL**, called **KL-UCRL** that uses modified confidence intervals on the transition kernel based on Kullback-Leibler divergence rather than $|| \cdot ||_1$ control on the error. The resulting algorithm is reported to work better in practice, although this is not reflected in the theoretical bounds. Farahmand (2011) introduced a metric for MDPs called the action-gap. This work is the closest in spirit to our approach. The action-gap captures the difficulty of distinguishing the optimal policy from near-optimal policies, and is complementary to the notion of hardness proposed here. However, the action-gap has mainly been used for planning, instead of learning, which is our main focus. In the discounted setting, several works have improved the bounds with respect to the number of states (Szita and Szepesvári, 2010) and the discount factor (Lattimore and Hutter, 2012). However, these analyses focus on worst case bounds that do not scale with the hardness of the MDP, missing an opportunity to help bridge the gap between theory and practice.

**Contributions** Our main contribution is a refined metric for the hardness of MDPs, that captures the observed "easiness" of common benchmark MDPs. To accomplish this we first introduce a norm induced by a distribution $\nu$, aka the *distribution*-norm. For functions $f$ with zero $\nu$-expectation, $||f||_\nu$ is the variance of $f$. We define the dual of this norm in Lemma 1, and then study its concentration properties in Theorem 1. This central result is of independent interest beyond its application in RL. More precisely, for a discrete probability measure $p$ and its empirical version $\widehat{p}_n$ built from $n$ i.i.d samples, we control $||p - \widehat{p}_n||_{\star,p}$ in $O((np_0)^{-1/2})$, where $p_0$ is the minimum mass of $p$ on its support. Second, we define a hardness measure for MDPs based on the distribution-norm. This measure captures stochasticity along the value function. This quantity is naturally small in MDPs that are nearly deterministic, but it can also be small in MDPs with highly stochastic transition kernels. For instance, this is the case when all states reachable from a state have the same value. We show that some common benchmark MDPs have small hardness measure. This illustrates that our proposed norm is a useful tool for the analysis and design of existing and future RL algorithms.

**Outline** In Section 2, we formalize the distribution-norm, and give intuition about the interplay with its dual. We compare to distribution-independent norms. Theorem 1 provides a concentration inequality for the dual of this norm, that is of independent interest beyond the MDP setting. Section 3 uses these insights to define a *problem-dependent* hardness metric for both undiscounted and discounted MDPs (Definition 2, Definition 1), that we call the *environmental norm*. Importantly, we show in section 3.2 that common benchmark MDPs have small environmental norm $C$ in this sense, and compare our bound to approaches bounding the *problem-free* $|| \cdot ||_1$ norm.

## 2 The distribution-norm and its dual

In Machine Learning (ML), norms often play a crucial role in obtaining performance bounds. One typical example is the following. Let $\mathcal{X}$ be a measurable space equipped with an unknown probability measure $\nu \in \mathcal{M}_1(\mathcal{X})$ with density $p$. Based on some procedure, an algorithm produces a candidate measure $\tilde{\nu} \in \mathcal{M}_1(\mathcal{X})$ with density $\tilde{p}$. One is then interested in the loss with respect to a continuous function $f$. It is natural to look at the mismatch between $\nu$ and $\tilde{\nu}$ on $f$. That is

$$(\nu - \tilde{\nu}, f) = \int_{\mathcal{X}} f(x)(\nu - \tilde{\nu})(dx) = \int_{\mathcal{X}} f(x)(p(x) - \tilde{p}(x))dx \,.$$

A typical bound on this quantity is obtained by applying a Hölder inequality to $f$ and $p - \tilde{p}$, which gives $(\nu - \tilde{\nu}, f) \leqslant ||p - \tilde{p}||_1 ||f||_\infty$. Assuming a bound is known for $||f||_\infty$, this inequality can be controlled with a bound on $||p - \tilde{p}||_1$. When $\mathcal{X}$ is finite and $\tilde{p}$ is the empirical distribution $\widehat{p}_n$ estimated from $n$ i.i.d. samples of $p$, results such as Weissman et al. (2003) can be applied to bound this term with high probability.

However, in this learning problem, what matters is not $f$ but the way $f$ behaves with respect to $\nu$. Thus, trying to capture the properties of $f$ via the distribution-free $||f||_\infty$ bound is not satisfactory. So we propose, instead, a norm $|| \cdot ||_\nu$ driven by $\nu$. Well-behaving $f$ will have small norm $||f||_\nu$, whereas badly-behaving $f$ will have large norm $||f||_\nu$. Every distribution has a natural norm asso-

ciated with it that measures the quadratic variations of $f$ with respect to $\nu$. This quantity is at the heart of many key results in mathematical statistics, and is formally defined by

$$||f||_\nu = \sqrt{\int_\mathcal{X} \left(f(x) - \mathbb{E}_\nu f\right)^2 \nu(dx)} \,. \tag{1}$$

To get a norm, we restrict $\mathcal{C}(\mathcal{X})$ to the space of continuous functions $\mathcal{E}_\nu = \{f \in \mathcal{C}(\mathcal{X}) : ||f||_\nu < \infty, \mathbf{supp}(\nu) \subset \mathbf{supp}(f), \mathbb{E}_\nu f = 0\}$. We then define the corresponding dual space in a standard way by $\mathcal{E}_\nu^\star = \{\mu : ||\mu||_{\star,\nu} < \infty\}$ where

$$||\mu||_{\star,\nu} = \sup_{f \in \mathcal{E}_\nu} \frac{\int_x f(x)\mu(dx)}{||f||_\nu} \,.$$

Note that for $f \in \mathcal{E}_\nu$, using the fact the $\nu(\mathcal{X}) = \tilde{\nu}(\mathcal{X}) = 1$ and that $x \to f(x) - \mathbb{E}_\nu f$ is a zero mean function, we immediately have

$$
\begin{aligned}
(\nu - \tilde{\nu}, f) &= (\nu - \tilde{\nu}, f - \mathbb{E}_\nu f) \\
&\leqslant ||p - \tilde{p}||_{\star,\nu}||f - \mathbb{E}_\nu f||_\nu \,.
\end{aligned}
\tag{2}
$$

The key difference with the generic Hölder inequality is that $|| \cdot ||_\nu$ is now capturing the behavior of $f$ with respect to $\nu$, as opposed to $|| \cdot ||_\infty$. Conceptually, using a quadratic norm instead of an L1 norm, as we do here, is analogous to moving from Hoeffding's inequality to Bernstein's inequality in the framework of concentration inequalities.

We are interested in situations where $||f||_\nu$ is much smaller than $||f||_\infty$. That is, $f$ is well-behaving with respect to $\nu$. In such cases, we can get an improved bound $||p - \tilde{p}||_{\star,\nu}||f - \mathbb{E}_\nu f||_\nu$ instead of the best possible generic bound $\inf_{c \in \mathbb{R}} ||p - \tilde{p}||_1 ||f - c||_\infty$.

Simply controlling either $||p - \tilde{p}||_{\star,\nu}$ (respectively $||p - \tilde{p}||_1$) or $||f||_\nu$ (respectively $||f||_\infty$) is not enough. What matters is the product of these quantities. For our choice of norm, we show that $||p - \tilde{p}||_{\star,\nu}$ concentrates at essentially the same speed as $||p - \tilde{p}||_1$, but $||f||_\infty$ is typically much larger than $||f||_\nu$ for the typical functions met in the analysis of MDPs. We do not claim that the norm defined in equation (1) is the best norm that leads to a minimal $||p - \tilde{p}||_{\star,\nu}||f - \mathbb{E}_\nu f||_\nu$, but we show that it is an interesting candidate.

We proceed in two steps. First, we design in Section 2 a concentration bound for $||p - \widehat{p}_n||_{\star,\nu}$ that is not much larger than the Weissman et al. (2003) bound on $||p - \widehat{p}_n||_1$. (Note that $||p - \widehat{p}_n||_{\star,\nu}$ must be larger than $||p - \widehat{p}_n||_1$ as it captures a refined property). Second, in Section 3, we consider RL in an MDP where $p$ represents the transition kernel of a station-action pair and $f$ represents the value function of the MDP for a policy. The value function and $p$ are strongly linked by construction, and the distribution-norm helps us capture their interplay. We show in Section 3.2 that common benchmark MDPs have optimal value functions with small $|| \cdot ||_\nu$ norm. This naturally introduces a new way to capture the hardness of MDPs, besides the diameter (Jaksch, 2010) or the span (Bartlett and Tewari, 2009). Our formal notion of MDP hardness is summarized in Definitions 1 and 2, for discounted and undiscounted MDPs, respectively.

## 2.1 A dual-norm concentration inequality

For convenience we consider a finite space $\mathcal{X} = \{1, \dots, S\}$ with $S$ points. We focus on the first term on the right hand side of (2), which corresponds to the dual norm when $\tilde{p} = \widehat{p}_n$ is the empirical mean built from $n$ i.i.d. samples from the distribution $\nu$. We denote by $p$ the probability vector corresponding to $\nu$. The following lemma, whose proof is in the supplementary material, provides a convenient way to compute the dual norm.

**Lemma 1** *Assume that $\mathcal{X} = \{1, \dots, S\}$, and, without loss of generality, that $\mathbf{supp}(p) = \{1, \dots, K\}$, with $K \leqslant S$. Then the following equality holds true*

$$||\widehat{p}_n - p||_{\star,p} = \sqrt{\sum_{s=1}^K \frac{\widehat{p}_{n,s}^2 - p_s^2}{p_s}} \,.$$

Now we provide a finite-sample bound on our proposed norm.

**Theorem 1 (Main result)** *Assume that* $\mathbf{supp}(p) = \{1, \ldots, K\}$, *with* $K \leqslant S$. *Then for all* $\delta \in (0,1)$, *with probability higher than* $1 - \delta$,

$$||\widehat{p}_n - p||_{\star,p} \;\leqslant\; \min\left\{ \sqrt{\frac{1}{p_{(K)}} - 1}, \sqrt{\frac{K-1}{n}} + 2\sqrt{\frac{(2n-1)\ln(1/\delta)}{n^2}\left(\frac{1}{p_{(K)}} - \frac{1}{p_{(1)}}\right)} \right\}, \quad (3)$$

*where* $p_{(K)}$ *is the smallest non zero component of* $p = (p_1, \ldots, p_S)$, *and* $p_{(1)}$ *the largest one.*

The proof follows an adaptation of Maurer and Pontil (2009) for empirical Bernstein bounds, and uses results for self-bounded functions from the same paper. This gives tighter bounds than naive concentration inequalities (Hoeffding, Bernstein, etc.). We indeed get a $O(n^{-1/2})$ scaling, whereas using simpler techniques would lead to a weak $O(n^{-1/4})$ scaling.

**Proof** We will apply Theorem 7 of Maurer and Pontil (2009). Using the notation of this theorem, we denote the sample by $\mathbf{X} = (X_1, \ldots, X_n)$ and the function we want to control by

$$\mathcal{V}(\mathbf{X}) = ||\widehat{p}_n - p||_{\star,p}^2\,.$$

We now introduce, for any $s \in \mathcal{S}$ the modified sample $\mathbf{X}_{i_0,s} = (X_1, \ldots, X_{i_0-1}, s, X_{i_0+1}, \ldots, X_n)$. We are interested in the quantity $\mathcal{V}(\mathbf{X}) - \mathcal{V}(\mathbf{X}_{i_0,s})$. To apply Theorem 7 of Maurer and Pontil (2009), we need to identify constants $a, b$ such that

$$\begin{cases} \forall i \in [n], \quad \mathcal{V}(\mathbf{X}) - \inf_{s \in \mathcal{S}} \mathcal{V}(\mathbf{X}_{i,s}) & \leqslant b \\ \sum_{i=1}^{n}\left(\mathcal{V}(\mathbf{X}) - \inf_{s \in \mathcal{S}} \mathcal{V}(\mathbf{X}_{i,s})\right)^2 & \leqslant a\mathcal{V}(\mathbf{X})\,. \end{cases}$$

The two following lemmas enable us to identify $a$ and $b$. They follow from simple algebra and are proved in Appendix A in the supplementary material.

**Lemma 2** $\mathcal{V}(\mathbf{X})$ *satisfies* $\mathbb{E}_p\left[\mathcal{V}(\mathbf{X})\right] = \frac{K-1}{n}$. *Moreover, for all* $i \in \{1, \ldots, n\}$ *we have that*

$$\mathcal{V}(\mathbf{X}) - \inf_{s \in \mathcal{S}} \mathcal{V}(\mathbf{X}_{i,s}) \leqslant b\,, \;\text{ where } b = \frac{2n-1}{n^2}\left(\frac{1}{p_{(K)}} - \frac{1}{p_{(1)}}\right).$$

**Lemma 3** $\mathcal{V}(\mathbf{X}) = ||\widehat{p}_n - p||_{\star,p}^2$ *satisfies*

$$\sum_{i=1}^{n}\left(\mathcal{V}(\mathbf{X}) - \inf_{s \in \mathcal{S}} \mathcal{V}(\mathbf{X}_{i,s})\right)^2 \leqslant 2b\mathcal{V}(\mathbf{X})\,.$$

Thus, we can choose $a = 2b$. By application of Theorem 7 of Maurer and Pontil (2009) to $\tilde{\mathcal{V}}(\mathbf{X}) = \mathcal{V}(\mathbf{X})/b$, we deduce that for all $\varepsilon > 0$,

$$\mathbb{P}\left[\tilde{\mathcal{V}}(\mathbf{X}) - \mathbb{E}\tilde{\mathcal{V}}(\mathbf{X}) > \varepsilon\right] \;\leqslant\; \exp\left(-\frac{\varepsilon^2}{4\mathbb{E}\tilde{\mathcal{V}}(\mathbf{X}) + 2\varepsilon}\right).$$

Plugging back in the definition of $\tilde{\mathcal{V}}(\mathbf{X})$, we obtain

$$\mathbb{P}\left[||\widehat{p}_n - p||_{\star,p}^2 > \frac{K-1}{n} + \varepsilon\right] \;\leqslant\; \exp\left(-\frac{\varepsilon^2/b}{4\frac{K-1}{n} + 2\varepsilon}\right).$$

After inverting this bound in $\varepsilon$ and using the fact that $\sqrt{a+b} \leqslant \sqrt{a} + \sqrt{b}$ for non-negative $a, b$, we deduce that for all $\delta \in (0,1)$, with probability higher than $1 - \delta$, then

$$\begin{aligned} ||\widehat{p}_n - p||_{\star,p}^2 &\leqslant \mathbb{E}\mathcal{V}(\mathbf{X}) + 2\sqrt{\mathbb{E}\mathcal{V}(\mathbf{X})b\ln(1/\delta)} + 2b\log(1/\delta) \\ &= \left(\sqrt{\mathbb{E}\mathcal{V}(\mathbf{X})} + \sqrt{b\ln(1/\delta)}\right)^2 + b\log(1/\delta)\,. \end{aligned}$$

Thus, we deduce from this inequality that

$$\begin{aligned} ||\widehat{p}_n - p||_{\star,p} &\leqslant \sqrt{\mathbb{E}\mathcal{V}(\mathbf{X})} + 2\sqrt{b\ln(1/\delta)} \\ &= \sqrt{\frac{K-1}{n}} + 2\sqrt{\frac{(2n-1)\ln(1/\delta)}{n^2}\left(\frac{1}{p_{(K)}} - \frac{1}{p_{(1)}}\right)}\,, \end{aligned}$$

which concludes the proof. We recover here a $O(n^{-1/2})$ behavior, more precisely a $O(p_{(K)}^{-1}n^{-1/2})$ scaling where $p_{(K)}$ is the smallest non zero probability mass of $p$. $\qquad\square$

# 3 Hardness measure in Reinforcement Learning using the distribution-norm

In this section, we apply the insights from Section 2 for the distribution-norm to learning in Markov Decision Processes (MDPs). We start by defining a formal notion of hardness $C$ for discounted MDPs and undiscounted MDPs with average reward, that we call the *environmental norm*. Then, we show in Section 3.2 that several benchmark MDPs have small environmental norm. In Section 3.1, we present a regret bound for a modification of **UCRL** whose regret scales with $C$, without having to know $C$ in advance.

**Definition 1 (Discounted MDP)** *Let $M = <\mathcal{S}, \mathcal{A}, r, p, \gamma>$ be a $\gamma$-discounted MDP, with reward function $r$ and transition kernel $p$. We denote $V^\pi$ the value function corresponding to a policy $\pi$ (Puterman, 1994). We define the environmental-value norm of policy $\pi$ in MDP $M$ by*

$$C_M^\pi = \max_{s,a \in \mathcal{S} \times \mathcal{A}} ||V^\pi||_{p(\cdot|s,a)} \, .$$

**Definition 2 (Undiscounted MDP)** *Let $M = <\mathcal{S}, \mathcal{A}, r, p>$ be an undiscounted MDP, with reward function $r$ and transition kernel $p$. We denote by $h^\pi$ the bias function for policy $\pi$ (Puterman, 1994; Jaksch, 2010). We define the environmental-value norm of policy $\pi$ in MDP $M$ by the quantity*

$$C_M^\pi = \max_{s,a \in \mathcal{S} \times \mathcal{A}} ||h^\pi||_{p(\cdot|s,a)} \, .$$

In the discounted setting with bounded rewards in $[0,1]$, $V^\pi \leqslant \frac{1}{1-\gamma}$ and thus $C_M^\pi \leqslant \frac{1}{1-\gamma}$ as well. In the undiscounted setting, then $||h^\pi||_{p(\cdot|s,a)} \leqslant \mathbf{span}(h^\pi)$, and thus $C_M^\pi \leqslant \mathbf{span}(h^\pi)$. We define the class of $C$-"hard" MDPs by $\mathfrak{M}_C = \left\{ M : C_M^{\pi^*} \leqslant C \right\}$. That is, the class of MDPs with optimal policy having a low environmental-value norm, or for short, *MDPs with low environmental norm*.

**Important note** It may be tempting to think that, since the above definition captures a notion of variance, an MDP that is very noisy will have a high environmental norm. However this reasoning is incorrect. The environmental norm of an MDP is not the variance of a roll-out trajectory, but rather captures the variations of the value (or the bias value) function with respect to the transition kernel. For example, consider a fully connected MDP with transition kernel that transits to every state uniformly at random, but with a constant reward function. In this trivial MDP, $C_M^\pi = 0$ for all policies $\pi$, even though the MDP is extremely noisy because the value function is constant. In general MDPs, the environmental norm depends on how varied the value function is at the possible next states and on the distribution over next states. Note also that we use the term hardness rather than complexity to avoid confusion with such concepts as Rademacher or VC complexity.

## 3.1 "Easy" MDPs and algorithms

In this section, we demonstrate how the dual norm (instead of the usual $|| \cdot ||_1$ norm) can lead to improved bounds for learning in MDPs with small environmental norm.

**Discounted MDPs** Due to space constraints, we only report one proposition that illustrates the kind of achievable results. Indeed, our goal is not to derive a modified version of each existing algorithm for the discounted scenario, but rather to instill the key idea of using a refined hardness measure when deriving the core lemmas underlying the analysis of previous (and future) algorithms.

The analysis of most RL algorithms for the discounted case uses a "simulation lemma" (Kearns and Singh, 2002); see also Strehl and Littman (2008) for a refined version. A simulation lemma bounds the error in the value function of running a policy planned on an estimated MDP in the MDP where the samples were taken from. This effectively controls the number of samples needed from each state-action pair to derive a near-optimal policy. The following result is a simulation lemma exploiting our proposed notion of hardness (the environmental norm).

**Proposition 1** *Let $M$ be a $\gamma$-discounted MDP with deterministic rewards. For a policy $\pi$, let us denote its corresponding value $V^\pi$. We denote by $p$ the transition kernel of $M$, and for convenience use the notation $p^\pi(s'|s)$ for $p(s'|s, \pi(s))$. Now, let $\widehat{p}$ be an estimate of the transition kernel such that $\max_{s \in \mathcal{S}} ||p^\pi(\cdot|s) - \widehat{p}^\pi(\cdot|s)||_{\star, p^\pi(\cdot|s)} \leqslant \varepsilon$ and let us denote $\widehat{V}^\pi$ its corresponding value in the MDP with kernel $\widehat{p}$. Then, the maximal expected error between the two values is bounded by*

$$\mathcal{E}_{rr}^\pi \overset{\text{def}}{=} \max_{s_0 \in \mathcal{S}} \left( \mathbb{E}_{p^\pi(\cdot|s_0)} \left[ V^\pi \right] - \mathbb{E}_{\widehat{p}^\pi(\cdot|s_0)} \left[ \widehat{V}^\pi \right] \right) \leqslant \frac{\varepsilon C^\pi}{1 - \gamma} \, ,$$

*where $C^\pi = \max_{s,a \in \mathcal{S} \times \mathcal{A}} ||V^\pi||_{p(\cdot|s,a)}$. In particular, for the optimal policy $\pi^\star$, then $C^{\pi^\star} \leqslant C$.*

To understand when this lemma results in smaller sample sizes, we need to compare to what one would get using the standard $|| \cdot ||_1$ decomposition, for an MDP with rewards in $[0, 1]$. If $\max_{s \in \mathcal{S}} ||p^{\pi}(\cdot|s) - \widehat{p}^{\pi}(\cdot|s)||_1 \leqslant \varepsilon'$, then one would get

$$\mathcal{E}_{rr}^{\pi} \leqslant \frac{\varepsilon \mathbf{span}(V^{\pi})}{1 - \gamma} \leqslant \frac{\varepsilon' V_{MAX}^*}{1 - \gamma} \leqslant \frac{\varepsilon'}{(1 - \gamma)^2} \, .$$

When, for example, $C$ is a bound with respect to all policies, this simulation lemma can be plugged directly into the analysis of R-MAX (Kakade, 2003) or MBIE (Strehl and Littman, 2008) to obtain a hardness-sensitive bound on the sample complexity. Now, in most analyses, one only needs to bound the hardness with respect to the optimal policy and to the optimistic/greedy policies actually used by the algorithm. For an optimal policy $\tilde{\pi}$ computed from an $(\varepsilon, \varepsilon')$-approximate model (see Lemma 4 for details), it is not difficult to show that $C^{\tilde{\pi}} \leqslant C^{\pi^\star} + (\varepsilon' C^{\pi^\star} + \varepsilon)/(1 - \gamma)$, which thus allows for a tighter analysis. We do not report further results here, to avoid distracting the reader from the main message of the paper, which is the introduction of a distribution-dependent hardness metric for MDPs. Likewise, we do not detail the steps that lead from this result to the various sample-complexity bounds one can find in the abundant literature on the topic, as it would not be more illuminating than Proposition 1.

**Undiscounted MDPs** In the undiscounted setting, with average reward criterion, it is natural to consider the **UCRL** algorithm from Jaksch (2010). We modify the definition of plausible MDPs used in the algorithm as follows: Using the same notations as that of Jaksch (2010), we replace the admissibility condition for a candidate transition kernel $\tilde{p}$ at the beginning of episode $k$ at time $t_k$

$$||\widehat{p}_k(\cdot|s, a) - \tilde{p}(\cdot|s, a)||_1 \leqslant \sqrt{\frac{14S \log(2At_k/\delta)}{\max\{1, N_k(s, a)\}}} \, ,$$

with the following condition involving the result of Theorem 1

$$||\widehat{p}_k(\cdot|s, a) - \tilde{p}(\cdot|s, a)||_{\star, \tilde{p}(\cdot|s, a)} \leqslant B_k(s, a) \overset{\text{def}}{=}$$

$$\min\left\{\sqrt{\frac{1}{p_0} - 1}, \sqrt{\frac{K - 1}{\max\{1, N_k(s, a)\}}} + 2\sqrt{\frac{(2N_k(s, a) - 1)\ln(t_k SA/\delta)}{\max\{1, N_k(s, a)\}^2}\left(\frac{1}{\tilde{p}_{(K)}} - \frac{1}{\tilde{p}_{(1)}}\right)}\right\}, (4)$$

where $\tilde{p}_{(K)}$ is the smallest non zero component of $\tilde{p}(\cdot|s, a)$, and $\tilde{p}_{(1)}$ the largest one, and $K$ is the size of the support of $\tilde{p}(\cdot|s, a)$. We here assume for simplicity that the transition kernel $p$ of the MDP always puts at least $p_0$ mass on each point of its support, and thus constraint an admissible kernel $\tilde{p}$ to satisfy the same condition. One restriction of the current (simple) analysis is that the algorithm needs to know a bound on $p_0$ in advance. We believe it is possible to remove such an assumption by estimating $p_0$ and taking care of the additional low probability event corresponding to the estimation error. As this comes at the price of a more complicated algorithm and analysis, we do not report this extension here for clarity. Note that the optimization problem corresponding to Extended Value Iteration with (4) can still be solved by optimizing over the simplex. We refer to Jaksch (2010) for implementation details. Naturally, similar modifications apply also to **REGAL** and other **UCRL** variants introduced in the MDP literature.

In order to assess the performance of the policy chosen by **UCRL** it is useful to show the following:
**Lemma 4** *Let $M$ and $\tilde{M}$ be two communicating MDPs over the same state-action space such that one is an $(\varepsilon, \varepsilon')$-approximation of the other in the sense that for all $s, a \; |r(s, a) - \tilde{r}(s, a)| \leqslant \varepsilon$ and $||\tilde{p}(\cdot|s, a) - p(\cdot|s, a)||_{\star, p(\cdot|s, a)} \leqslant \varepsilon'$. Let $\rho^\star(M)$ denotes the average value function of $M$. Then*

$$||\rho^\star(M) - \rho^\star(\tilde{M})||_p \leqslant \varepsilon' \min\{C_M, C_{\tilde{M}}\} + \varepsilon \, .$$

Lemma 4 is a simple adaptation from Ortner et al. (2014). We now provide a bound on the regret of this modified **UCRL** algorithm. The regret bound turns out to be a bit better than **UCRL** in the case of an MDP $M \in \mathfrak{M}_C$ with a small $C$.

**Proposition 2** *Let us consider a finite-state MDP with $S$ state, low environmental norm ($M \in \mathfrak{M}_C$) and diameter $D$. Assume moreover that the transition kernel that always puts at least $p_0$ mass on each point of its support. Then, the modified **UCRL** algorithm run with condition (4) is such that for all $\delta$, with probability higher than $1 - \delta$, for all $T$, the regret after $T$ steps is bounded by*

$$\mathfrak{R}_T = O\left(\left[DC\sqrt{SA}\left(\sqrt{\frac{\log(TSA/\delta)}{p_0}} + \sqrt{S}\right) + D\right]\sqrt{\frac{T}{p_0}\log(TSA/\delta)}\right).$$

The regret bound for the original **UCRL** from Jaksch (2010) scales as $O\left(DS\sqrt{AT\log(TSA/\delta)}\right)$. Since we used some crude upper bounds in parts of the proof of Proposition 2, we believe the right scaling for the bound of Proposition 2 is $O\left(C\sqrt{\frac{TSA}{p_0}\log(TSA/\delta)}\right)$. The cruder factors come from some second order terms that we controlled trivially to avoid technical and not very illuminating considerations. What matters here is that $C$ appears as a factor of the leading term. Indeed proposition 2 is mostly here for illustration purpose of what one can achieve, and improving on the other terms is technical and goes beyond the scope of this paper. Comparing the two regret bounds, the result of Proposition 2 provides a qualitative improvement over the result of Jaksch (2010) whenever $C < D\sqrt{S}p_0$ (respectively $C < \sqrt{S}p_0$) for the conjectured (resp. current) result.

**Note**. The modified **UCRL** algorithm does not need to know the environmental norm $C$ of the MDP in advance. It only appears in the analysis and in the final regret bound. This property is similar to that of **UCRL** with respect to the diameter $D$.

## 3.2 The hardness of benchmarks MDPs

In this section, we consider the hardness of a set of MDPs that have appeared in past literature. Table 3.2 summarizes the results for six MDPs that were chosen to be both representative of typical finite-states MDPs but also cover a diverse range of tasks. These MDPs are also significant in the sense that good solutions for them have been learned with far fewer samples then suggested by existing theoretical bounds. The metrics we report include the number of states $S$, the number of actions $A$, the maximum of $V^\star$ (denoted $V^*_{\text{MAX}}$), the span of $V^*$, the $C^{\pi^*}_M$, and $p_0 = \min_{s\in\mathcal{S},a\in\mathcal{A}} \min_{s'\in\mathbf{supp}(p(\cdot|s,a)} p(s'|s,a)$, that is the minimum non-zero probability mass given by the transition kernel of the MDP. While we cannot compute the hardness for all policies, the hardness with respect to $\pi^*$ is significant because it indicates how hard it is to learn the value function $V^*$ of the optimal policy. Notice that $C^{\pi^*}_M$ is significantly smaller than both $V^*_{\text{MAX}}$ and $\mathbf{span}(V^*)$ in all the MDPs. This suggests that a model accurately representing the optimal value function can be derived with a small number of samples (and a bound based on $\|\cdot\|_1 V^*_{\text{MAX}}$ is overly conservative).

| MDP | $S$ | $A$ | $V^*_{\text{MAX}}$ | $\mathbf{Span}(V^*)$ | $C^{\pi^*}_M$ | $p_0$ |
|---|---|---|---|---|---|---|
| bottleneck McGovern and Barto (2001) | 231 | 4 | 19.999 | 19.999 | 0.526 | 0.1 |
| red herring Hester and Stone (2009) | 121 | 4 | 17.999 | 17.999 | 4.707 | 0.1 |
| taxi † Dieterich (1998) | 500 | 6 | 7.333 | 0.885 | 0.055 | 0.043 |
| inventory † Mankowitz et al. (2014) | 101 | 2 | 19.266 | 0.963 | 0.263 | $< 10^{-3}$ |
| mountain car † ⋆ ⋄ Sutton and Barto (1998) | 150 | 3 | 19.999 | 19.999 | 1.296 | 0.322 |
| pinball † ⋆ ⋄ Konidaris and Barto (2009) | 2304 | 5 | 19.999 | 19.991 | 0.059 | $< 10^{-3}$ |

Table 1: MDPs marked with a † indicate that the true MDP was not available and so it was estimated from samples. We estimated these MDPs with $10,000$ samples from each state-action pair. MDPs marked with a ⋆ indicate that the original MDP is deterministic and therefore we added noise to the transition dynamics. For the Mountain Car problem, we added a small amount of noise to the vehicle's velocity during each step ($pos_{t+1} = pos_t + vel_t(1 + X)$ where $X$ is a random variable with equally probable events $\{-vel_{MAX}, 0, vel_{MAX}\}$). For the pinball domain we added noise similar to Tamar et al. (2013). MDPs marked with a ⋄ were discretized to create a finite state MDP. The rewards of all MDPs were normalized to $[0, 1]$ and discount factor $\gamma = 0.95$ was used.

To understand the environmental-value norm of near-optimal policies $\pi$ in an MDP, we ran policy iteration on each of the benchmark MDPs from Table 3.2 for 100 iterations (see supplementary material for further details). We computed the environmental-value norm of all encountered policies and selected the policy $\pi$ with maximal norm and its corresponding worst case distribution. Figure 1 compares the Weissman et al. (2003) bound $\times V_{\text{MAX}}$ to the bound given by Theorem 1 $\times C^\pi_M$ as the number of samples increases. This is indeed the comparison of this products that matters for the learning regret, rather than that of one or the other factor only. In each MDP, we see an order of magnitude improvement by exploiting the distribution-norm. This is particularly significant because the Weissman et al. (2003) bound is quite close to the behavior observed in experiments. The result in Figure 1 strengthens support for our theoretical findings, suggesting that bounds based on the distribution-norm scale with the MDP's hardness.

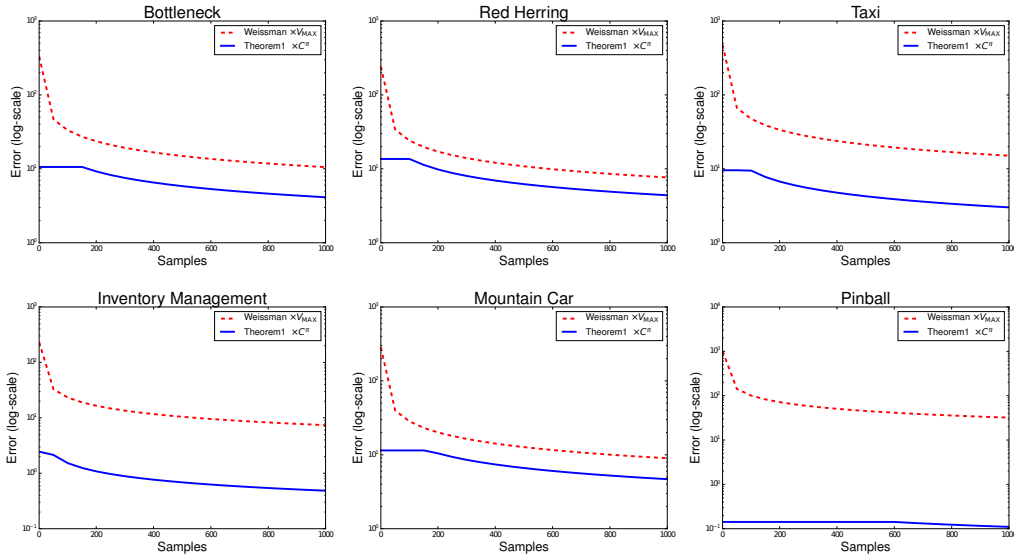

Figure 1: Comparison of the Weissman et al. (2003) bound times $V_{\mathrm{MAX}}$ to (3) of Theorem 1 times $C_M^\pi$ in the benchmark MDPs. In each MDP, we selected the policy $\pi$ (from the policies encountered during policy iteration) that gave the largest $C^\pi$ and the worst next state distribution for our bound. In each MDP, the improvement with the distribution-norm is an order of magnitude (or more) better than using the distribution-free Weissman et al. (2003) bound.

## 4    Discussion and conclusion

In the early days of learning theory, sample independent quantities such as the VC-dimension and later the Rademacher complexity were used to derive generalization bounds for supervised learning. Later on, data dependent bounds (empirical VC or empirical Rademacher) replaced these quantities to obtain better bounds. In a similar spirit, we proposed the first analysis in RL where instead of considering generic a-priori bounds one can use stronger MDP-specific bounds. Similarly to the supervised learning, where generalization bounds have been used to drive model selection algorithms and structural risk minimization, our proposed distribution dependent norm suggests a similar approach in solving RL problems. Although we do not claim to close the gap between theoretical and empirical bounds, this paper opens an interesting direction of research towards this goal, and achieves a significant first step. It inspires at least a modification of the whole family of UCRL-based algorithms, and could potentially benefit also to others fundamental problems in RL such as basis-function adaptation or model selection, but efficient implementation should not be overlooked.

We choose a natural weighted $L_2$ norm induced by a distribution, due to its simplicity of interpretation and showed several benchmark MDPs have low hardness. A natural question is how much benefit can be obtained by studying other $L_p$ or Orlicz distribution-norms? Further, one may wish to create other distribution dependent norms that emphasize certain areas of the state space in order to better capture desired (or undesired) phenomena. This is left for future work.

In the analysis we basically showed how to adapt existing algorithms to use the new distribution dependent hardness measure. We believe this is only the beginning of what is possible, and that new algorithms will be developed to best utilize distribution dependent norms in MDPs.

**Acknowledgements** This work was supported by the European Community's Seventh Framework Programme (FP7/2007-2013) under grant agreement 306638 (SUPREL) and the Technion.

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
