[Supplementary Material]

# Supplementary material to the paper: " How hard is my MDP?" The distribution-norm to the rescue.

## A Proofs regarding the dual norm

In this section, we provide the detailed proofs of the results corresponding to the dual norm $||\cdot||_{\star,p}$, in the case when $\mathcal{X}$ is a discrete space.

### A.1 Proof of Lemma 1

**Lemma 1** *When $\mathcal{X} = \{1,\ldots,S\}$ and $\mathbf{supp}(p) = \{1,\ldots,K\}$ with $K \leqslant S$, then the following equality holds true*

$$||\widehat{p}_n - p||_{\star,p} = \sup_{f\in\mathcal{E}_p:||f||_p=1} \int_x f(x)\widehat{p}_n(x) = \sqrt{\sum_{s=1}^K \frac{\widehat{p}_{n,s}^2 - p_s^2}{p_s}}\,.$$

**Proof:** The first equality is by definition. Introducing two Lagrangian parameters $\alpha$ and $\beta$ corresponding to the two equality constraints $||f||_p = 1$ and $\mathbb{E}_p f = 0$, an optimal solution $f^\star$ satisfies that $\alpha^\star(||f^\star||_p^2 - 1) = 0$ and $\beta^\star \mathbb{E}_p f = 0$. We write $p = (p_1,\ldots,p_K,0,\ldots,0)^\top \in \Delta_S$ and then it holds by the KKT conditions that

$$\forall s \in \{1,\ldots,K\}, \quad \widehat{p}_{n,s} - p_s = 2\alpha^\star f_s^\star p_s + \beta^\star p_s\,; \qquad \sum_{s=1}^K f_s^{\star 2} p_s = 1 \qquad \sum_{s=1}^K f_s^\star p_s = 0\,.$$

Thus, we deduce on the one hand that

$$\forall s \in \{1,\ldots,K\}, \quad f_s^\star = \frac{\widehat{p}_{n,s} - (1+\beta^\star)p_s}{2\alpha^\star p_s}\,, \quad \text{with } \sum_{s=1}^K \widehat{p}_{n,s} = (1+\beta^\star)\,,$$

but we must have also $\sum_{s=1}^K \widehat{p}_{n,s} = 1$, thus $\beta = 0$. On the other hand, we have

$$\alpha^\star = \frac{1}{2}\sqrt{\sum_{s=1}^K \frac{(\widehat{p}_{n,s} - p_s)^2}{p_s}}\,.$$

Plugging-in back the expression of $f^\star$ in the definition of the dual norm, we deduce that

$$||\widehat{p}_n - p||_{\star,p} = \sum_{s=1}^K \frac{\left(\widehat{p}_{n,s} - p_s\right)\widehat{p}_{n,s'}/p_s}{\sqrt{\sum_{s=1}^K \left(\widehat{p}_{n,s} - p_s\right)^2/p_s}}\,.$$

Let us simplify this expression. We have on the one hand

$$\sum_{s=1}^K \left(\widehat{p}_{n,s} - p_s\right)\widehat{p}_{n,s'}/p_s = \left(\sum_{s=1}^K \frac{\widehat{p}_{n,s}^2}{p_s}\right) - 1\,,$$

and on the other hand, it holds that

$$\sum_{s=1}^K \left(\widehat{p}_{n,s} - p_s\right)^2/p_s = \sum_{s=1}^K \frac{\widehat{p}_{n,s}^2}{p_s} + p_s - 2\widehat{p}_{n,s} = \left(\sum_{s=1}^K \frac{\widehat{p}_{n,s}^2}{p_s}\right) - 1\,.$$

Thus, we deduce the following simplified expression

$$||\widehat{p}_n - p||_{\star,p} = \sqrt{\sum_{s=1}^K \frac{\widehat{p}_{n,s}^2 - p_s^2}{p_s}}\,.$$

$\square$

## A.2 Proof of Lemma 2

**Lemma 2** $\mathcal{V}(\mathbf{X})$ *satisfies* $\mathbb{E}_p\left[\mathcal{V}(\mathbf{X})\right] = \frac{K-1}{n}$. *Moreover, for all* $i \in \{1, \ldots, n\}$ *we have that*

$$\mathcal{V}(\mathbf{X}) - \inf_{s \in \mathcal{S}} \mathcal{V}(\mathbf{X}_{i,s}) \leqslant b\,, \text{ where } b = \frac{2n-1}{n^2}\left(\frac{1}{p_{(K)}} - \frac{1}{p_{(1)}}\right).$$

**Proof:** We start by decomposing $||\widehat{p}_n - p||^2_{\star,2,p}$ in terms of the random variables $\{X_i\}_{i=\in[n]}$. We get that

$$
\begin{aligned}
\mathcal{V}(\mathbf{X}) = ||\widehat{p}_n - p||^2_{\star,p} &= \left(\sum_{s=1}^{K}\sum_{i=1}^{n}\frac{\mathbb{I}\{X_i = s\}}{n^2 p_s} + \sum_{i=1}^{n}\sum_{i'\neq i=1}^{n}\frac{\mathbb{I}\{X_i = s\}\mathbb{I}\{X_{i'} = s\}}{n^2 p_s}\right) - 1 \\
&= \sum_{i=1}^{n}\left(\frac{1}{n^2 p_{X_i}} + \sum_{i'\neq i=1}^{n}\frac{\mathbb{I}\{X_i = X_{i'}\}}{n^2 p_{X_i}}\right) - 1\,.
\end{aligned}
\tag{5}
$$

Note also that with this expression, we derive easily

$$\mathbb{E}_p\left[\mathcal{V}(\mathbf{X})\right] = \sum_{s=1}^{K}\left(\frac{1}{n} + p_s\frac{n(n-1)}{n^2}\right) - 1 = \frac{K-1}{n}\,.$$

For $s \in \mathcal{S}$, we have, from (5)

$$\mathcal{V}(\mathbf{X}) - \mathcal{V}(\mathbf{X}_{i,s}) = \frac{1}{n^2 p_{X_{i_0}}} - \frac{1}{n^2 p_s} + 2\sum_{i\neq i_0=1}^{n}\left(\frac{\mathbb{I}\{X_i = X_{i_0}\}}{n^2 p_{X_{i_0}}} - \frac{\mathbb{I}\{X_i = s\}}{n^2 p_s}\right).$$

Thus, if we introduce $p_{(1)}$ to be the largest component of $p$ and $p_{(K)}$ its smallest non-0 component, we deduce that

$$
\begin{aligned}
\mathcal{V}(\mathbf{X}) - \min_{s \in \mathcal{S}}\mathcal{V}(\mathbf{X}_{i,s}) &\leqslant \frac{1}{n^2}\left(\frac{1}{p_{(K)}} - \frac{1}{p_{(1)}} + 2\frac{(n-1)}{p_{(K)}} - 2\frac{(n-1)}{p_{(1)}}\right) \\
&= \frac{2n-1}{n^2}\left(\frac{1}{p_{(K)}} - \frac{1}{p_{(1)}}\right).
\end{aligned}
$$

We thus set $b = \frac{2n-1}{n^2}\left(\frac{1}{p_{(K)}} - \frac{1}{p_{(1)}}\right)$. $\qquad\square$

## A.3 Proof of Lemma 3

**Lemma 3** *The quantity* $\mathcal{V}(\mathbf{X}) = ||\widehat{p}_n - p||^2_{\star,p}$ *satisfies*

$$\sum_{i=1}^{n}\left(\mathcal{V}(\mathbf{X}) - \inf_{s \in \mathcal{X}}\mathcal{V}(\mathbf{X}_{i,s})\right)^2 \leqslant 2bV(\mathbf{X})\,.$$

**Proof:** On the one hand, we have

$$\mathcal{V}(\mathbf{X}) = \sum_{i=1}^{n}\left(\frac{1}{n^2 p_{X_i}} + \sum_{i'\neq i=1}^{n}\frac{\mathbb{I}\{X_i = X_{i'}\}}{n^2 p_{X_i}}\right) - 1\,.$$

Now, on the other hand, since $\mathcal{V}(\mathbf{X}) - \mathcal{V}(\mathbf{X}_{i,s_i}) \geqslant 0$, the quantity we want to control satisfies

$$
\begin{aligned}
\sum_{i=1}^{n} \Big( \mathcal{V}(\mathbf{X}) - \mathcal{V}(\mathbf{X}_{i,s_i}) \Big)^2 &= \sum_{i=1}^{n} \left[ \frac{1}{n^2 p_{X_{i_0}}} - \frac{1}{n^2 p_{s_i}} + 2 \sum_{i \neq i_0 = 1}^{n} \left( \frac{\mathbb{I}\{X_i = X_{i_0}\}}{n^2 p_{X_{i_0}}} - \frac{\mathbb{I}\{X_i = s_i\}}{n^2 p_{s_i}} \right) \right]^2 \\
&\leqslant b \sum_{i=1}^{n} \left[ \frac{1}{n^2 p_{X_{i_0}}} + 2 \sum_{i \neq i_0 = 1}^{n} \frac{\mathbb{I}\{X_i = X_{i_0}\}}{n^2 p_{X_{i_0}}} - \frac{2(n-1)}{n^2 p_{(1)}} - \frac{1}{n^2 p_{(1)}} \right] \\
&\leqslant b \left( \mathcal{V}(\mathbf{X}) + 1 + \sum_{i \neq i_0 = 1}^{n} \frac{\mathbb{I}\{X_i = X_{i_0}\}}{n^2 p_{X_{i_0}}} - \frac{2}{p_{(1)}} \right) \\
&\leqslant b \left( \mathcal{V}(\mathbf{X}) + \sum_{i \neq i_0 = 1}^{n} \frac{\mathbb{I}\{X_i = X_{i_0}\}}{n^2 p_{X_{i_0}}} - 1 \right) \\
&\leqslant 2 b \mathcal{V}(\mathbf{X}) \, .
\end{aligned}
$$

$\square$

## B   Discounted MDP

In this section, we provide the detailed proofs of the results that correspond to the performance analysis of algorithms that use the $|| \cdot ||_{\star, p}$ confidence bounds instead of $|| \cdot ||_1$ bounds in the case of discounted MDPs.

**Proposition 1:** *Let $M$ be a $\gamma$-discounted MDP with deterministic rewards, and $\pi$ be a policy, with corresponding value $V^\pi$. We denote by $p$ the transition kernel of $M$, and for convenience use the notation $p^\pi(s'|s)$ for $p(s'|s, \pi(s))$. Now, let $\widehat{p}$ be some estimate transition kernel such that $\max_{s \in \mathcal{S}} ||p^\pi(\cdot|s) - \widehat{p}^\pi(\cdot|s)||_{\star, p^\pi(\cdot|s)} \leqslant \varepsilon$ and let us denote $\widehat{V}^\pi$ its corresponding value in the MDP with kernel $\widehat{p}$. Then, the maximal expected error between the two values is bounded by*

$$
\mathcal{E}_{rr}^\pi \overset{\text{def}}{=} \max_{s_0 \in \mathcal{S}} \Big( \mathbb{E}_{p^\pi(\cdot|s_0)} \big[ V^\pi \big] - \mathbb{E}_{\widehat{p}^\pi(\cdot|s_0)} \big[ \widehat{V}^\pi \big] \Big) \leqslant \frac{\varepsilon C^\pi}{1 - \gamma} \, .
$$

**Proof:** Simple algebra shows that

$$
\begin{aligned}
\mathcal{E}_{rr}^\pi &= \max_{s_0 \in \mathcal{S}} \sum_{s \in \mathcal{S}} \big( V^\pi(s) p^\pi(s|s_0) - V^\pi(s) \widehat{p}^\pi(s|s_0) \big) + \sum_{s \in \mathcal{S}} \big( V^\pi(s) \widehat{p}^\pi(s|s_0) - \widehat{V}^\pi(s) \widehat{p}^\pi(s|s_0) \big) \\
&= \max_{s_0 \in \mathcal{S}} \sum_{s \in \mathcal{S}} V^\pi(s) \big( p^\pi(s|s_0) - \widehat{p}^\pi(s|s_0) \big) + \sum_{s \in \mathcal{S}} \widehat{p}^\pi(s|s_0) \big( V^\pi(s) - \widehat{V}^\pi(s) \big) \, .
\end{aligned}
$$

Now, on the one hand, we have by property of the dual norm, and definition of $\varepsilon$ and $C$ that

$$
\begin{aligned}
\sum_{s \in \mathcal{S}} V^\pi(s) \big( p^\pi(s|s_0) - \widehat{p}^\pi(s|s_0) \big) &\leqslant ||p^\pi(\cdot|s_0) - \widehat{p}^\pi(\cdot|s_0)||_{\star, p^\pi(\cdot|s_0)} ||V^\pi - \sum_{s \in \mathcal{S}} V^\pi(s) p^\pi(s|s_0)||_{p^\pi(\cdot|s_0)} \\
&\leqslant \varepsilon C \, .
\end{aligned}
$$

On the other hand, we use one step of the Bellman equation together with the fact that the reward is deterministic to deduce that

$$
\begin{aligned}
\sum_{s \in \mathcal{S}} \widehat{p}^\pi(s|s_0) \big( V^\pi(s) - \widehat{V}^\pi(s) \big) &= \gamma \sum_{s \in \mathcal{S}} \widehat{p}^\pi(s|s_0) \Big( \sum_{s' \in \mathcal{S}} \widehat{V}^\pi(s') p^\pi(s'|s) - \widehat{V}^\pi(s'|s) \widehat{p}^\pi(s'|s) \Big) \\
&\leqslant \gamma \Big( \sum_{s \in \mathcal{S}} \widehat{p}^\pi(s|s_0) \Big) \max_{s \in \mathcal{S}} \Big( \sum_{s' \in \mathcal{S}} \widehat{V}^\pi(s') p^\pi(s'|s) - \widehat{V}^\pi(s'|s) \widehat{p}^\pi(s'|s) \Big) \\
&= \gamma \mathcal{E}_{rr}^\pi \, ,
\end{aligned}
$$

where the last equality is because $\sum_{s \in \mathcal{S}} \widehat{p}^\pi(s|s_0) = 1$. Thus, we obtain $\mathcal{E}_{rr}^\pi \leqslant \varepsilon C + \gamma \mathcal{E}_{rr}^\pi$, that is

$$
\mathcal{E}_{rr}^\pi \leqslant \frac{\varepsilon C}{1 - \gamma} \, .
$$

$\square$

# C   Undiscounted MDP

In this section, we provide detailed proofs of the results that correspond to the regret analysis of the modified **UCRL** algorithm that uses the $|| \cdot ||_{\star,p}$ confidence bounds instead of $|| \cdot ||_1$ bounds. We reused the notations from Jaksch (2010).

## C.1   Proof of Proposition 2

**Proposition 2** *Let us consider a finite-state MDP with $S$ states, low kernel variance $M \in \mathfrak{M}_C$ and diameter $D$. Assume moreover that the transition kernel that always puts at least $p_0$ mass on each point of its support. Then, the modified* **UCRL** *algorithm run with condition* (4) *is such that for all $\delta$, with probability higher than $1 - \delta$, for all $T$, the regret after $T$ steps is bounded by*

$$
\mathfrak{R}_T \;=\; O\left( \left[ DC\sqrt{SA}\Big( \sqrt{\frac{\log(TSA/\delta)}{p_0}} + \sqrt{S} \Big) + D \right] \sqrt{\frac{T}{p_0} \log(TSA/\delta)} \right),
$$

We reuse most of the analysis of **UCRL**, and only change the steps corresponding to the use of the modified confidence intervals (4) for admissible transition kernels. Since the original proof of **UCRL** is quite long, we decided not to re-derive the whole proof in a self-contained way. The corresponding modifications would have been lost in the details. Instead, we refer precisely to the steps that need to modified in the original proof, and provide the corresponding modifications below. We also use the same notations as that of Jaksch (2010) for clarity.

**Proof:** The proof follows exactly the same steps as the regret proof given by Jaksch (2010) for **UCRL**, up to two differences. More precisely, the very same steps hold until Section 4.3.2 of Jaksch (2010). In this step, we need to update equation (17) and deal with $\mathbf{v}_k(\tilde{\mathbf{P}}_k - \mathbf{P}_k)\mathbf{w}_k$. Since the rows of both $\tilde{\mathbf{P}}_k$ and $\mathbf{P}_k$ sum to 1, this quantity is invariant under a translation of $\mathbf{w}_k$ by a constant. Remember that $w_k$ is defined from the value $u_i$ computed by the Extended Value Iteration algorithm in episode $k$ by

$$
w_k(s) = u_i(s) + \frac{\min_s u_i(s) + \max_s u_i(s)}{2} \, .
$$

For our purpose, we now define for each $s \in \mathcal{S}$, first $w_{k,s}(s') = u_i(s') - \mathbb{E}_{p(\cdot|s,\tilde{\pi}_k(s))}[u_i]$ and then $\tilde{w}_{k,s}(s') = u_i(s') - \mathbb{E}_{\tilde{p}_k(\cdot|s,\tilde{\pi}_k(s))}[u_i]$. We then derive a replacement for (17) from Jaksch (2010)

$$
\begin{aligned}
\mathbf{v}_k(\tilde{\mathbf{P}}_k - \mathbf{P}_k)\mathbf{w}_k &= \sum_s \sum_{s'} v_k(s, \tilde{\pi}_k(s)) \cdot \Big( \tilde{p}_k(s'|s, \tilde{\pi}_k(s)) - p(s'|s, \tilde{\pi}_k(s)) \Big) u_i(s') \\
&\leqslant \sum_s v_k(s, \tilde{\pi}_k(s)) \Big( ||\tilde{p}_k(\cdot|s, \tilde{\pi}_k(s)) - \widehat{p}(\cdot|s, \tilde{\pi}_k(s))||_{\star, \tilde{p}_k(\cdot|s, \tilde{\pi}_k(s))} \cdot ||\tilde{w}_{k,s}||_{\tilde{p}_k(\cdot|s, \tilde{\pi}_k(s))} \\
&\qquad + ||\widehat{p}(\cdot|s, \tilde{\pi}_k(s)) - p(\cdot|s, \tilde{\pi}_k(s))||_{\star, p(\cdot|s, \tilde{\pi}_k(s))} \cdot ||w_{k,s}||_{p(\cdot|s, \tilde{\pi}_k(s))} \Big) \\
&\leqslant \sum_s v_k(s, \tilde{\pi}_k(s)) B_k\big(s, \tilde{\pi}_k(s)\big) \Big( ||w_{k,s}||_{p(\cdot|s, \tilde{\pi}_k(s))} + ||\tilde{w}_{k,s}||_{\tilde{p}(\cdot|s, \tilde{\pi}_k(s))} \Big) . \qquad (6)
\end{aligned}
$$

At this point, we now relate $||\tilde{w}_k||_{\tilde{p}(\cdot|s, \tilde{\pi}_k(s))} = ||u_i||_{\tilde{p}(\cdot|s, \tilde{\pi}_k(s))}$ to the definition of $C$. In Jaksch (2010), one could simply use the diameter of the MDP. Here, we need to work a little more. The following lemma establishes a relationship between $||\tilde{w}_k||_{\tilde{p}(\cdot|s\tilde{\pi}_k(s))}$ and $C$.

**Lemma 5** *Provided that the MDP $M$ is admissible in episode $k$, then the approximated optimistic value computed by Extended Value Iteration satisfies that*

$$
||u_i||_{\tilde{p}(\cdot|s,a)} \;\leqslant\; ||h||_{\tilde{p}(\cdot|s, \tilde{\pi}_k(s))} + 2D(B_k C + B_k^r) + \frac{D}{\sqrt{t_k}} \, ,
$$

*where $B_k = \max_{s,a} B_k(s, a)$ and $B_k^r = \max_{s,a} \min\{1, \sqrt{\frac{7 \log(2SAt_k/\delta)}{\max\{1, N_k(s,a)\}}}\} \leqslant 1$.*

We then relate $||\tilde{w}_k||_{p(\cdot|s,\tilde{\pi}_k(s))} = ||u_i||_{p(\cdot|s,\tilde{\pi}_k(s))}$ to $C$ as well, and $||h||_{\tilde{p}(\cdot|s,\tilde{\pi}_k(s))}$ to $||h||_{p(\cdot|s,\tilde{\pi}_k(s))} \leqslant C$ thanks to the following lemma

**Lemma 6** *Provided that the MDP $M$ is admissible in episode $k$, then it holds that*

$$||h||^2_{p(\cdot|s,a)} \leqslant ||h||^2_{\tilde{p}(\cdot|s,a)} + 2D^2 B_k(s,a) \,,$$

*where $D$ is the diameter of the true MDP. Further, the same holds for all $f$ with* **span**$(f) \leqslant D$.

Thanks to these lemmas, we deduce that, provided that the true MDP is admissible in episode $k$, then

$$
\begin{aligned}
\mathbf{v}_k(\tilde{\mathbf{P}}_k - \mathbf{P}_k)\mathbf{w}_k &\leqslant \sum_s v_k(s, \tilde{\pi}_k(s)) B_k(s, \tilde{\pi}_k(s))\Big( 2C + 4B_k CD + 3\sqrt{2}D\sqrt{B_k(s, \tilde{\pi}_k(s))} \\
&\quad + 4D + \frac{2D}{\sqrt{t_k}} \Big).
\end{aligned}
$$

Note that this a crude bound, since $2D(B_k C + B_k^r) + \frac{D}{\sqrt{t_k}}$ is actually a second order term. We believe it is possible to take advantage of this with a much trickier analysis (by controlling $B_k^r$ and $B_k$ for all $t$).

The second term in section 4.3.2 that needs to be controlled is $X_t = \langle p(\cdot|s_t, a_t) - e_{s_{t+1}}, w_{k(t)}\rangle \mathbb{I}\{M \in \mathcal{M}_{k(t)}\}$, where $M$ is the true MDP and $\mathcal{M}_{k(t)}$ denotes the set of plausible MDPs computed in episode $k(t)$.

**Lemma 7** *We have the property that if $M$ is admissible in episode $k = k(t)$, then*

$$|X_t| \leqslant \frac{1}{\sqrt{2p_0}} \min\left\{ D, C + 2C\Big(\frac{1}{p_0} - 1\Big)^{1/2} + D\Big(\frac{1}{p_0} - 1\Big)^{1/4} + \sqrt{2} + 1 \right\}.$$

From this point on, one can use the same next steps of the analysis by Jaksch (2010) and conclude similarly to their result. Denoting by $m$ the number of episodes as in Jaksch (2010), equation (18) in Jaksch (2010) is replaced with

$$
\begin{aligned}
&\sum_{k=1}^{m} \mathbf{v}_k(\mathbf{P}_k - \mathbf{I})\mathbf{w}_k \mathbb{I}\{M \in \mathcal{M}_k\} \\
&\leqslant \sum_{t=1}^{T} X_t + mD \\
&\leqslant D\sqrt{\frac{5T}{4p_0}\log\left(\frac{8T}{\delta}\right)} + DSA\log_2\left(\frac{8T}{SA}\right),
\end{aligned}
$$

with probability higher than $1 - \frac{\delta}{12T^{5/4}}$. We then deal with equation (17) in Jaksch (2010). First, we bound $B_k(s,a)$ by

$$
\begin{aligned}
B_k(s,a) &\leqslant 2\sqrt{\frac{(2N_k(s,a) - 1)\ln(t_k SA/\delta)}{\max\{1, N_k(s,a)\}^2}\left(\frac{1}{p_0} - 1\right)} + \sqrt{\frac{K-1}{\max\{1, N_k(s,a)\}}} \\
&\leqslant \left(2\sqrt{\frac{2\log(t_k SA/\delta)}{p_0}} + \sqrt{K-1}\right)\frac{1}{\sqrt{\max\{1, N_k(s,a)\}}}.
\end{aligned}
$$

Then, we deduce that equation (17) in Jaksch (2010) is replaced with

$$
\begin{aligned}
\mathbf{v}_k(\tilde{\mathbf{P}}_k - \mathbf{P}_k)\mathbf{w}_k &\leqslant \left(2\sqrt{\frac{2\log(t_k SA/\delta)}{p_0}} + \sqrt{K-1}\right)\Big[2\big(C + 2B_k CD + 2D\big) \\
&\quad \times \sum_{s,a} \frac{v_k(s,a)}{\sqrt{\max\{1, N_k(s,a)\}}} + O\Big(\sum_{s,a} \frac{v_k(s,a)}{N_k^{3/4}(s,a)}\frac{D}{p_0^{1/4}}\Big)\Big].
\end{aligned}
$$

As a result, we obtain a bound on the sum of the regret in each episode $\Delta_k$, summing over all episodes $k \leqslant m$ such that $M$ is admissible. We get with probability higher than $1 - \frac{\delta}{12T^{5/4}}$ that

$$\sum_{k=1}^{m} \Delta_k \mathbb{I}\{M \in \mathcal{M}_k\} \leqslant$$

$$2\Big(C + 2B_k CD + 2D\Big)\Big(2\sqrt{\frac{2\log(TSA/\delta)}{p_0}} + \sqrt{K-1}\Big)\sum_{k=1}^{m}\sum_{s,a}\frac{v_k(s,a)}{\sqrt{N_k(s,a)}}$$

$$+D\sqrt{\frac{5T}{4p_0}\log\Big(\frac{8T}{\delta}\Big)} + DSA\log_2\Big(\frac{8T}{SA}\Big) + O\Big(\Big(\frac{T}{p_0}\Big)^{1/4}DSA\log_2\Big(\frac{8T}{SA}\Big)\Big)$$

$$+\Big(\sqrt{14\log(\frac{2SAT}{\delta})} + 2\Big)\sum_{k=1}^{m}\sum_{s,a}\frac{v_k(s,a)}{\sqrt{N_k(s,a)}}\ .$$

Let us now introduce he notation $\tilde{C} = C + 2DC\Big(2\sqrt{\frac{2\log(TSA/\delta)}{p_0}} + \sqrt{K-1}\Big) + 2D$. Using the same simplifying arguments as in Jaksch (2010), we can replace equation (21) in Jaksch (2010) with

$$\sum_{k=1}^{m} \Delta_k \mathbb{I}\{M \in \mathcal{M}_k\} \leqslant \left[\Big[\frac{4\sqrt{2}\tilde{C}}{\sqrt{p_0}} + 2\sqrt{14}\Big]\sqrt{\log(2TSA/\delta)} + 2\tilde{C}\sqrt{S-1}\right]\Big(\sqrt{2}+1\Big)\sqrt{SAT}$$

$$+D\sqrt{\frac{5T\log(8T/\delta)}{4p_0}} + O\Big(\Big(\frac{T}{p_0}\Big)^{1/4}DSA\log_2\Big(\frac{8T}{SA}\Big)\Big)\ .$$

The regret of the modified **UCRL** algorithm is thus given by the following bound, with probability higher than $1 - \frac{\delta}{12T^{5/4}} - \frac{\delta}{12T^{5/4}} - \frac{\delta}{12T^{5/4}}$.

$$\mathfrak{R}_T \leqslant \sqrt{\frac{5}{8}T\log(\frac{8T}{\delta})} + \sqrt{T} + D\sqrt{\frac{5T}{4p_0}\log\Big(\frac{8T}{\delta}\Big)} + O\Big(\Big(\frac{T}{p_0}\Big)^{1/4}DSA\log_2\Big(\frac{8T}{SA}\Big)\Big)$$

$$\left[\Big[\frac{4\sqrt{2}\tilde{C}}{\sqrt{p_0}} + 2\sqrt{14}\Big]\sqrt{\log(TSA/\delta)} + 2\tilde{C}\sqrt{S-1}\right]\Big(\sqrt{2}+1\Big)\sqrt{SAT}\ .$$

Since $\sum_{T=2}^{\infty}\frac{\delta}{4T^{5/4}} < \delta$, we deduce that with probability higher than $1 - \delta$, uniformly for all $T$, then $\mathfrak{R}_T = O\Big(\big(\tilde{C}\sqrt{SA} + D\big)\sqrt{\frac{T}{p_0}\log(TSA/\delta)} + \tilde{C}S\sqrt{AT}\Big)$. $\qquad\square$

### C.2 Proof of Lemma 5

**Lemma 5** *Provided that the MDP $M$ is admissible in episode $k$, then the approximated optimistic value computed by Extended Value Iteration satisfies that*

$$||u_i||_{\tilde{p}(\cdot|s,\tilde{\pi}_k(s))} \leqslant ||\tilde{h}||_{\tilde{p}(\cdot|s,\tilde{\pi}_k(s))} + \frac{D}{\sqrt{t_k}}$$

$$\leqslant ||h||_{\tilde{p}(\cdot|s,\tilde{\pi}_k(s))} + 2(B_k C + B_k^r)D + \frac{D}{\sqrt{t_k}}\ ,$$

*where $B_k = \max_{s,a} B_k(s,a)$ and $B_k^r = \max_{s,a}\sqrt{\frac{7\log(2SAt_k/\delta)}{\max\{1,N_k(s,a)\}}}$.*

**Proof:** Let us denote for convenience $\tilde{p}$ for $\tilde{p}(\cdot|s,\tilde{\pi}_k(s))$. We first relate $||u_i||_{\tilde{p}}^2$ to $||\tilde{h}^{\tilde{\pi}}||_{\tilde{p}}^2$.

First, following our analysis one can easily derive the following adaptation of Lemma 8 from Ortner et al. (2014).

**Lemma 8** *Consider a communicating MDP $M = (S, A, r, p)$, and another MDP $\tilde{M} = (S, A, \tilde{r}, \tilde{p})$ over the same state-action space which is an $(\varepsilon, \varepsilon')$-approximation of $M$, in the sense that for all*

$s, a \, |r(s,a) - \tilde{r}(s,a)| \leqslant \varepsilon|$ *and* $||\tilde{p}(\cdot|s,a) - p(\cdot|s,a)||_{\star,p(\cdot|s,a)} \leqslant \varepsilon'$. *Assume that an optimal policy* $\pi^\star$ *for* $M$ *is performed on* $\tilde{M}$ *for* $\ell$ *steps, and let* $\tilde{v}^\star(s)$ *be the number of times state* $s$ *is visited state among these* $\ell$ *steps. Then*

$$\ell\rho^\star(M) - \sum_s \tilde{v}^\star(s)\tilde{r}(s,\pi^\star(s)) < \ell\big(\varepsilon'C + \varepsilon\big) + D + D\sqrt{\frac{\ell\log(1/\delta)}{p_0}}\,.$$

An immediate corollary, that is the analogue of Lemma 9 from Ortner et al. (2014) is the following

**Lemma 9** *Let* $M$, $\tilde{M}$ *be two communicating MDPs over the same state-action space such that one is an* $(\varepsilon, \varepsilon')$-*approximation of the other. Then,*

$$|\rho^\star(M) - \rho^\star(\tilde{M})| \leqslant \varepsilon'\min\{C_M, C_{\tilde{M}}\} + \varepsilon\,.$$

Now, we use the fact that the Poisson equation that defines the optimal bias function $h$ in $M$ and $\tilde{h}$ in $\tilde{M}$ in involves $\rho, r, p$, such that

$$\rho^\star + h(s) = \max_{a \in \mathcal{A}}\left[r(s,a) + \sum_{s' \in \mathcal{S}} p(s'|s,a)h(s')\right].$$

Thus, we deduce from Lemma 9 a similar result for the span

**Lemma 10** *Let* $M$, $\tilde{M}$ *be two communicating MDPs over the same state-action space such that one is an* $(\varepsilon, \varepsilon')$-*approximation of the other. Then,*

$$||h(M) - h(\tilde{M})||_p \leqslant 2\big(\varepsilon'\min\{C_M, C_{\tilde{M}}\} + \varepsilon\big)\min\{D_M, D_{\tilde{M}}\}\,.$$

The proof follows by using the Poisson equation for $h$ and $\tilde{h}$, then using the $\varepsilon$ approximation of $r$, the $\varepsilon$ approximation of $p$ that gives a term $\varepsilon\min\{C_M, C_{\tilde{M}}\}$, and the approximation of $\rho$ that gives the last $\varepsilon(\min\{C_M, C_{\tilde{M}}\} + 1)$. We also use that $h$ and $\tilde{h}$ are defined up to a constant. Finally, one needs to propagate the approximation error, which adds a factor $D$.

Indeed, by the Poisson equation, writing $h = h(M)$ and $\tilde{h} = h(\tilde{M})$, it holds that

$$\begin{aligned}\tilde{h}(s) &= \max_{a \in \mathcal{A}}\left[\sum_{s' \in \mathcal{S}} \tilde{p}(s'|s,a)\tilde{h}(s') + \tilde{r}(s,a)\right] - \rho(\tilde{M})\\[2mm] &= \max_{a \in \mathcal{A}}\left[\sum_{s' \in \mathcal{S}} p(s'|s,a)h(s') + r(s,a) + \sum_{s' \in \mathcal{S}} p(s'|s,a)(\tilde{h}(s') - h(s'))\right.\\[2mm] &\quad\left. + \sum_{s' \in \mathcal{S}} (\tilde{p}(s'|s,a) - p(s'|s,a))\tilde{h}(s') + \big(\tilde{r}(s,a) - r(s,a)\big)\right]\\[2mm] &\quad - \rho(M) + \big(\rho(M) - \rho(\tilde{M})\big)\,.\end{aligned}$$

Thus, we deduce that

$$|\tilde{h}(s) - h(s)| \leqslant |\rho(M) - \rho(\tilde{M})| + \varepsilon + \varepsilon'C_{\tilde{M}} + |\max_{a \in \mathcal{A}}\mathbb{E}_{p(\cdot|s,a)}(\tilde{h} - h)|\,.$$

Now, since $h$ and $\tilde{h}$ are defined up to a constant, it is always possible to make sure that $\max_{a \in \mathcal{A}}\mathbb{E}_{p(\cdot|s,a)}(\tilde{h} - h)$ for one state $s$. For the other states, we need to propagate the error bound. Since the diameter of the MDP is less than $D$, then we deduce that for all $s \in \mathcal{S}$

$$\begin{aligned}|\tilde{h}(s) - h(s)| &\leqslant \left(|\rho(M) - \rho(\tilde{M})| + \varepsilon + \varepsilon'\min\{C_M, C_{\tilde{M}}\}\right)D\\[2mm] &\leqslant 2\big(\varepsilon'\min\{C_M, C_{\tilde{M}}\} + \varepsilon)D\,,\end{aligned}$$

where we applied the result of Lemma 9. We conclude by symmetry.

We then apply this to the optimistic MDP, and get that $\varepsilon' = \max_{s,a} B_k(s,a)$ and $\varepsilon = \max_{s,a} B_k^r(s,a)$. Finally, in order to go from $u_i$ to $h$, we use the fact that $u_i$ satisfies an approximate Poisson equation, up to an error term that is controlled by $\frac{1}{\sqrt{t_k}}$ by equation (13) from Jaksch (2010). After propagation, this gives a $\frac{D}{\sqrt{t_k}}$ term. $\qquad\square$

## C.3 Proof of Lemma 6

**Lemma 6** *P*rovided that the MDP $M$ is admissible in episode $k$, then it holds that

$$||h||^2_{\tilde{p}(\cdot|s,a)} \leqslant ||h||^2_{p(\cdot|s,a)} + 2D^2 B_k(s,a),$$

where $D$ is the diameter of the true MDP, for any $h$ such that **span**$(h) \leqslant D$.

**Proof:** The proof is in two steps. First, using the short-hand notation $p = p(\cdot|s,a)$ and $\tilde{p} = \tilde{p}(\cdot|s,a)$, it holds that

$$
\begin{aligned}
||h||^2_p - ||h||^2_{\tilde{p}} &= \sum_{s'\in\mathcal{S}}(h(s') - \sum_{s''}h(s'')p_{s''})^2 p_{s'} - \sum_{s'\in\mathcal{S}}(h(s') - \sum_{s''}h(s'')\tilde{p}_{s''})^2 \tilde{p}_{s'}\\
&= \sum_{s'\in\mathcal{S}} h^2(s')\Big(p_{s'} - \tilde{p}_{s'}\Big) + \Big(\sum_{s'\in\mathcal{S}} h(s')(\tilde{p}_{s'} - p_{s'})\Big)\Big(\sum_{s'\in\mathcal{S}} h(s')(\tilde{p}_{s'} + p_{s'})\Big).
\end{aligned}
$$

Now, since both $||\cdot||^2_{\tilde{p}}$ and $||\cdot||^2_p$ are invariant if we translate the operand by a constant $c$, let us replace $h$ with $h - \mathbb{E}_{\tilde{p}}[h]$. In that case, we get

$$
\begin{aligned}
||h||^2_p - ||h||^2_{\tilde{p}} &= \sum_{s'\in\mathcal{S}}(h(s') - \mathbb{E}_{\tilde{p}}[h])^2\Big(p_{s'} - \tilde{p}_{s'}\Big) - \Big(\sum_{s'\in\mathcal{S}}(h(s') - \mathbb{E}_{\tilde{p}}[h])p_{s'}\Big)^2\\
&\leqslant \sum_{s'\in\mathcal{S}}(h(s') - \mathbb{E}_{\tilde{p}}[h])^2\Big(p_{s'} - \widehat{p}_{n,s'}\Big) + \sum_{s'\in\mathcal{S}}(h(s') - \mathbb{E}_{\tilde{p}}[h])^2\Big(\widehat{p}_{n,s'} - \tilde{p}_{s'}\Big)\\
&\leqslant ||(h(\cdot) - \mathbb{E}_{\tilde{p}}[h])^2||_p ||p - \widehat{p}_n||_{\star,p} + ||(h(\cdot) - \mathbb{E}_{\tilde{p}}[h])^2||_{\tilde{p}}||\widehat{p}_n - \tilde{p}||_{\star,\tilde{p}}.
\end{aligned}
$$

Now, we use the fact that $||(h(\cdot) - \mathbb{E}_{\tilde{p}}[h])^2||_q \leqslant$ **span**$(h)^2$, for $q = p$ and $q = \tilde{p}$, and then that **span**$(h)$ is upper bounded by the diameter $D$ of the true MDP. This is proved by a similar argument to that in Jaksch (2010), since we consider the same extended-action MDP. Thus, we deduce the bound

$$||h||^2_p \leqslant ||h||^2_{\tilde{p}} + D^2\Big(||p - \widehat{p}_n||_{\star,p} + ||\widehat{p}_n - \tilde{p}||_{\star,\tilde{p}}\Big).$$

$\square$

## C.4 Proof of Lemma 7

**Lemma 7** *L*et $X_t = \langle p(\cdot|s_t,a_t) - e_{s_{t+1}}, w_{k(t)}\rangle \mathbb{I}\{M \in \mathcal{M}_{k(t)}\}$. We have the property that if $M \in \mathcal{M}_{k(t)}$ (that is, the true MDP $M$ is admissible in episode $k = k(t)$), then

$$|X_t| \leqslant \frac{1}{\sqrt{2p_0}}\min\left\{D, C + 2C\Big(\frac{1}{p_0} - 1\Big)^{1/2} + D\Big(\frac{1}{p_0} - 1\Big)^{1/4} + \sqrt{2} + 1\right\}.$$

**Proof:** Indeed, $X_t$ satisfies, if $M \in \mathcal{M}_{k(t)}$

$$
\begin{aligned}
|X_t| &= \big|\langle p(\cdot|s_t,a_t) - e_{s_{t+1}}, w_{k(t)} - \mathbb{E}_{p(\cdot|s_t,a_t)}w_{k(t)}\rangle\big|\\
&= \big|\langle e_{s_{t+1}}, w_{k(t)} - \mathbb{E}_{p(\cdot|s_t,a_t)}w_{k(t)}\rangle\big|\\
&\leqslant ||e_{s_{t+1}}||_{\star,\tilde{p}(\cdot|s_t,a_t)}||w_{k(t)}||_{\tilde{p}(\cdot|s_t,a_t)}\\
&= ||e_{s_{t+1}}||_{\star,\tilde{p}(\cdot|s_t,a_t)}||u_i||_{\tilde{p}(\cdot|s_t,a_t)}
\end{aligned}
$$

Now, we deduce, from the following rewriting

$$||e_{s_{t+1}}||_{\star,\tilde{p}(\cdot|s_t,a_t)} = \sup\{f(s_{t+1}) : \sum_{s\in\mathcal{S}}f(s)\tilde{p}(s|s_t,a_t) = 0 \text{ and } \sum_{s\in\mathcal{S}}f^2(s)\tilde{p}(s|s_t,a_t) = 1\},$$

that we must have $f(s_{t+1}) \leqslant \frac{1}{\sqrt{2}\sqrt{\tilde{p}(s_{t+1}|s_t,a_t)}}$. Thus, using the assumption that either $p(s|s_t,a_t) > p_0$ or $p(s|s_t,a_t) = 0$ for all $s$, that $\tilde{p}$ must satisfy the same constraint, and using the result of Lemma 5, we deduce that (since we must have $p(s_{t+1}|s_t,a_t) > 0$)

$$
\begin{aligned}
|X_t| &\leqslant \frac{C + 2(B_k C + 1) + \frac{1}{\sqrt{t_k}} + 2D\sqrt{B_k(s_t,a_t)}}{\sqrt{2p_0}} \\
&= \frac{C(1 + 2B_k)}{\sqrt{2p_0}} + \frac{D}{\sqrt{2p_0}}\sqrt{B_k(s_t,a_t)} + \sqrt{\frac{2}{p_0}} + \frac{1}{\sqrt{2p_0 t_k}} \,.
\end{aligned}
$$

Now, we need a deterministic upper bound in order to be able to apply the result from Azuma's inequality. Thus, we use that $B_k(s_t,a_t) \leqslant \sqrt{\frac{1}{p_0} - 1}$, and get that

$$
|X_t| \leqslant \frac{1}{\sqrt{2p_0}}\left(C + 2C\left(\frac{1}{p_0} - 1\right)^{1/2} + D\left(\frac{1}{p_0} - 1\right)^{1/4} + \sqrt{2} + 1\right).
$$

$\square$

## D  Additional Experimental Details & Results

**Normalizing & Discounting:** For all the benchmark MDPs, we normalized the reward functions so that all rewards were within the range $[0,1]$. The inventory management task (Mankowitz et al., 2014) was originally a cost minization problem, so we negated the rewards before normalization to obtain a maximization problem. For all MDPs, we used a discount factor $\gamma = 0.95$.

**State Discretization:** The Mountain Car task and the Pinball domain both have continuous state-spaces. Thus, we needed to discretize them to obtain a finite-state MDP. States for the Mountain Car task are described by a position and velocity. The discretization used for the Mountain Car task was a grid where the cars position was divided into 15 bins and the velocity was devided into 10 bins. States for the Pinball domain are described by a 2-dimensional position and velocity. The discretization for the Pinball domain was 12 bins for the $x$-coordinate, 12 bins for the $y$-coordinate, 4 bins for the $x$-velocity, and 4 bins for the $y$-velocity.

**Policy Iteration:** For each benchmark MDP, we executed policy iteration for 100 iterations. Initial policies were generated by randomly selecting actions for each state according to a uniform distribution. During each iteration, we evaluated the current policy by executing the policy evaluation algorithm for 500 iterations.