[Reviews · NeurIPS 2014]

Submitted by Assigned_Reviewer_10

Note to authors: As I stated in my original review, though I think it could
be improved upon (and what paper couldn't?), I believe this paper should
be accepted. My apologies for not making that more clear. I have edited
my review below in an attempt to clarify some of my comments that were
perhaps not stated carefully enough originally.

---Summary of Paper---

The paper presents a novel measure of "hardness" for MDPs (based on a
novel concentration inequality) which essentially measures the
stochasticity of the *value* of the next state. Implications for
performance bounds for RL algorithms are sketched out. Empirically the
new measure is demonstrated to be tighter than previously known
indicators of MDP hardness.

---Quality---

The paper appears to be technically sound. The main results are
theoretical in nature and are well-supported.

Due to space constraints, Section 3.1 comes across as rushed and
incomplete. I understand the authors' excitement about the potentially
broad implications of their result, but unfortunately there is simply
not enough room to fully roll those implications out. As a result,
the paper currently presents some key implications (the new
simulation lemma, the modified UCRL bound), but leaves a lot of
follow through as exercise for the reader (see lines 288-292,
311-314, 315-317, 338-341). I certainly couldn't expect the authors
to fill in all those details in the available space, and I agree that some
of what they hint at is straightforward given sufficiently intimate
familiarity with the relevant literature. Nevertheless, in my opinion,
the presentation of this section leaves the reader feeling unresolved,
and, depending on how much their intuition overlaps with the authors,
perhaps unsure of whether the various paths the authors point to will
really be as smooth as they suggest. I wonder if the paper might have
been better served by thoroughly and concretely rolling out one
impactful consequence with hints in other directions, rather than the
sample platter of incomplete offshoots that is presented here.

The empirical results are helpful and evocative, and do support the
case that the new measure is tighter than other known measures. On the
other hand, they stop short of demonstrating a more concrete
connection between the bounds given and learning
performance. A more thorough evaluation might have measured if the
distinctions made by this quantity (some examples have notably higher C
than others) are reflected in practice, or included some examples thought
to be more difficult to show that the measure responds accordingly. Also
a variation on UCRL is presented with an improved regret bound due to
the result in this paper, but whether this also results in better
performance in practice is not investigated. The authors rightly point
out in their response that they make no claim to improve practical
performance, and I suppose therefore they are not beholden to demonstrate
practical impact. Nevertheless, this is the question we ultimately care
about, is it not? So it seems like it would have been worthwhile to
investigate/report, regardless of the outcome.

---Clarity---

I found the paper to be by and large clear, readable, and
well-written. I appreciated the authors' efforts to continually
connect technical statements/results with high level ideas and goals.

Continuing on my comments above, I will say that I feel the clarity of
Section 3 is harmed by the sheer number of points to be made. Space
constraints being what they are, this section moves at a rapid clip
and, as mentioned, leaves a lot as "exercise for the reader." I fear that
readers who are not as intimately familiar with the surrounding theoretical
context as the authors may not so readily leap to the conclusions that
the authors intend to communicate.

A couple of minor suggestions:

- On lines 19-21 the abstract contains a very nice intuitive example
of when the new bound can be much tighter. That example does appear
in the main text, but not until page 5, and even then it is
expressed in more technical terms. I'd recommend adding this
intuition to the introduction as well. I think it is useful for the
reader to have it in their head as they work through the results,
and the abstract should not be relied upon for the flow of the
paper.

- I'd humbly recommend that the authors give their measure a
name. Just writing this review it became clear how cumbersome it is
to discuss a nameless quantity. It may also have a positive impact
on readers' retention/understanding to have something more compact
and more evocative way to refer to this thing than C^\pi_M.

---Originality---

As far as I know the main results are novel and I agree with the
authors that their bound likely represents a new level of granularity
in formally characterizing the "hardness" of learning MDPs. The work
feels sufficiently contextualized in the relevant literature.

---Significance---

I think there is clear theoretical/scientific significance to the
result. The authors are correct to point out the often wide gap
between theoretical guarantees and practical experience. If we are
ever to gain a formal understanding of the phenomena we actually
observe (and if theoretical guarantees are ever going to be meaningful
to practitioners) results like this one, that examine problems at a
far finer granularity, will be required. It's not clear that this
result closes this gap, but I think it is compellingly demonstrated
that it takes a step in the right direction.

At the moment the practical significance of the result is unclear. The
paper does show that modifications to algorithms may be suggested by
the result, but it was not investigated whether the modification had
an impact in practice.

I do agree with the authors that there is very real *potential* for
this result to have wide-reaching consequences both for the analysis
of existing algorithms and for the development of new ones. Though in
some ways I feel the paper could benefit from being retooled, in the end,
I think that the potential for significant follow-up results overrides my
concerns about presentation.
Summary: The paper is clearly presented and technically sound. I feel that it
suffers from trying to accomplish too much in limited space, but the
main result is clear and, in my opinion, has a lot of potential to
inspire follow-up work. For that reason I recommend that this paper be
accepted.

Submitted by Assigned_Reviewer_19

The paper proposes the distribution norm, the standard variation of the value function with respect to the distribution, as a measure of hardness for learning a MDP. A concentration inequality for the dual of the distribution-norm is also provided showing that it behaves reasonably. Several common benchmark problems are shown to have low hardness when measured using the distribution norm.

The paper is clear and the idea is original as far as I know. The distribution norm is analogous to the variance and is intriguing. The concentration result is nice. For learning, Proposition 2 provides regret bound for UCRL in terms of the distribution norm. This provides improvement when \sqrt{S}p_0 is larger than C -- this is nice but not particularly compelling.

The experimental evidence is also not particularly compelling -- \sqrt{S}p_0 is larger than C for some of the MDPs in section 3.2 but not others. It would have been nice to see measurements on the correlation between C and the sample size required to learn to see if the measure would be useful for tasks such as model selection.

Overall, I like the ideas and I think there is enough in the paper for the conference but I think the learning and experimental results are currently not compelling.

Summary: Strengths: Interesting proposed measure of harness for learning MDPs based on both the value function as well as the distribution. May lead to interesting further work on improved theoretical results for learning and practical algorithms.

Weaknesses: Theoretical result on learning and experimental results are not yet compelling.

Submitted by Assigned_Reviewer_26

The paper proposes new "empirical" sampling bounds for MDPs; the bounds are much tighter but can be only computed based on a known value function. The main idea is to refine the existing sampling bounds using the interaction of the transition probabilities and the value function. The new empirical bounds are an order of magnitude smaller than previous bounds.

The authors describe a new take on bounding the sample complexity of an MDP. While the bounds cannot be computed apriori, they do represent a new tool in understanding the inherent difficulty of the problem. I believe these result are important.

The paper is well written. In terms of organization, I think it would be preferable to shorten Section 2 include a bit more of details on the proofs of the main results (Proposition 1 & 2) in the paper.

The results are technically correct as far as I can tell and the experimental evaluation is compelling.

Minor comments:
1) There are many types of "Undiscounted MDPs", please specify if it is average reward, or total reward, etc ...
2) Page 6, line 277: lemma -> proposition
3) Page 7, line 339: what does "we believe that" mean? There appears to be a quite significant discrepancy between Proposition 2 and the "believed" scaling, mostly for the term log(T S A).

Summary: An interesting paper with useful and important results for the sample-based reinforcement learning community.
Author Feedback
Author rebuttal: We want to thank all the reviewers for their positive and constructive comments. They will help us polish the final version of the paper.
We will add more detail to the discussion of Theorem 1, Proposition 1 and 2, and make precise the comments that may seem overly speculative.

The following provides more specific comments:

REVIEWER 10:

Sketchy discussion:
Section 3 explains the implications of Theorem 1 in detail with precise propositions. All related quantities are defined in the paper and the proof is given in the supplementary material. Without a precise example/line numbers, it is difficult to understand what you refer to when mentioning sketchy discussion.

Dividing an "old bound" by 10 does not give a theoretically valid upper bound.
There is a gap between empirical performance and the state-of-the-art theoretical bounds. Our contribution provides a bound that is tighter than previous theoretical bounds for a large class of problems.
Algorithmic implications and experimental improvements are interesting but may distract the reader from our main message.

Section 3 will be clarified in depth. We will modify the paper based on your minor suggestions as well.

We do not claim to close the gap between theoretical and empirical bounds. We believe this paper opens an interesting direction of research towards this goal, and achieves a significant first step.

Our results inspire a modification of UCRL, and thus the whole family of UCRL-based algorithms. This part is not speculative. Whether one can get efficient implementation of those and achieve better performance *in practice* than their original counterpart is not clear as it depends on the problem. We do show that one can get an improved performance upper-bound for one algorithm.

We will polish the camera-ready version of the paper, improve readability and preciseness in the parts that you suggested are less clear.
This only requires minor changes as there is no need to alter the main message of the paper. We don't think this justifies rejection.

REVIEWER 19:

Some MDPs are indeed simple and some are hard. Thus incorporating C into the bound will not always make performance bounds tighter. However, we argue that C will tighten the bounds in many practical MDPs.
You mention that "[i]t would have been nice to see measurements on the correlation between C and the sample size required to learn". We considered adding a figure similar to the one you suggest. However, the notion of when an algorithm has "learned" is not directly compatible with the on-line regret formulation discussed in section 3, thus this kind of figure might confuse the reader. Instead, we opted to provide Figure 1, which emphasizes the error bound as a function of the number of samples. Figure 1 in conjunction with Table 1 provides the reader with more insight about how C influences learning. Still, we are willing to add some figures in the supplementary material.

REVIEWER 26:

We provide a problem-dependent complexity measure. Such a quantity --by definition-- depends on the MDP. Examples of problem-dependent quantities include the diameter, the span of the optimal value function, and action gaps.

The complexity bound does *not* need to be known in advance by the modified UCRL algorithm. This is similar to UCRL with respect to the diameter. Thus not knowing it is not necessarily a problem. Of course, knowing an upper bound can also help.

Thank you for your minor comments. We will incorporate your suggestions into the final version of the paper.

Line 339: We used some crude upper bounds in parts of the proof of Proposition 2 for simplicity. Tightening these bounds is possible but at the price of more technical considerations that do not seem illuminating for the conference version of the paper. Actually Proposition 2 is mainly here for illustration. Our goal is not to get the tightest possible bound, but to demonstrate the potential of incorporating C into a regret bound.